# Meta Learning to Bridge Vision and Language Models for Multimodal Few-Shot Learning

**Ivona Najdenkoska[1], Xiantong Zhen[1,2], Marcel Worring[1]**
[1]AIM Lab, University of Amsterdam, [2]Inception Institute of Artificial Intelligence
`{i.najdenkoska, x.zhen, m.worring} @uva.nl`

## Abstract

Multimodal few-shot learning is challenging due to the large domain gap between vision and language modalities. Existing methods are trying to communicate visual concepts as prompts to frozen language models, but rely on hand-engineered task induction to reduce the hypothesis space. To make the whole process learnable, we introduce a multimodal meta-learning approach. Specifically, our approach decomposes the training of the model into a set of related multimodal few-shot tasks. We define a meta-mapper network, acting as a meta-learner, to efficiently bridge frozen large-scale vision and language models and leverage their already learned capacity. By updating the learnable parameters only of the meta-mapper, it learns to accrue shared meta-knowledge among these tasks. Thus, it can rapidly adapt to newly presented samples with only a few gradient updates. Importantly, it induces the task in a completely data-driven manner, with no need for a hand-engineered task induction. We evaluate our approach on recently proposed multimodal few-shot benchmarks, measuring how rapidly the model can bind novel visual concepts to words and answer visual questions by observing only a limited set of labeled examples. The experimental results show that our meta-learning approach outperforms the baseline across multiple datasets and various training settings while being computationally more efficient.

## 1 Introduction

Learning quickly by observing a few examples in a multimodal environment is an integral part of human intelligence (Schmidhuber, 1987; Bengio et al., 1991). Yet, it is quite challenging for current vision and language models to deal with a limited labeled space, when performing multimodal few-shot learning (Tsimpoukelli et al., 2021; Alayrac et al., 2022). On the contrary, language-only models have already flourished over the past years, especially when being transferred to a limited labeled space (Brown et al., 2020; Perez et al., 2021; Min et al., 2022), as a result of the large-scale pre-training and huge model capacity. Such advances in natural language processing inspired similar efforts in the vision domain, yielding large vision models with impressive few-shot and zero-shot image classification capabilities (Radford et al., 2021; Zhai et al., 2021; Jia et al., 2021). However, due to the large domain gap between vision and language modalities, it is non-trivial to directly embody few-shot capabilities into multimodal settings, which is the main motivation of this work.

One of the main challenges for bridging this gap is finding a proper mechanism for communicating visual concepts to a language model, by accruing shared knowledge from sequences of multimodal tasks. The Frozen model (Tsimpoukelli et al., 2021) is the first multimodal few-shot learner which tackles this challenge by taking inspiration from language models able to do in-context learning (Brown et al., 2020). This requires a task induction provided as a sentence followed by context data samples, to reduce the hypothesis space for open-ended image interpretation. This might not be a shortcoming for simpler tasks, like binary decisions; however it becomes an obstacle for more complicated tasks (Li & Liang, 2021), since the task induction has to be engineered each time. We hypothesize that the task can be induced from the data itself in a completely learnable manner.

Meta-learning or *learning to learn* (Schmidhuber, 1987; Thrun & Pratt, 2012; Andrychowicz et al., 2016) comes as a natural solution to any few-shot settings. While it has been extensively studied in unimodal settings, particularly for few-shot image classification (Vinyals et al., 2016; Ravi &

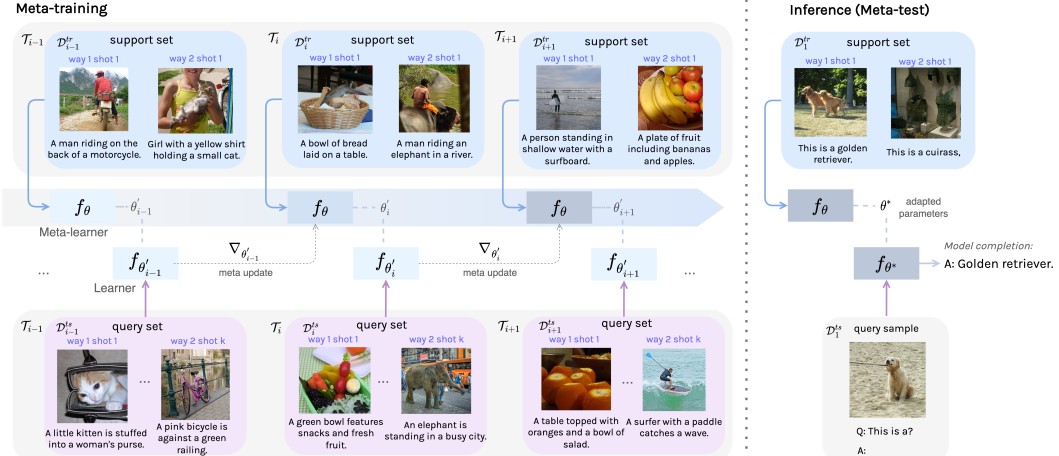

Figure 1: Multimodal few-shot meta-learning task for an example of a 2-way 1-shot setting, with two categories (ways) present in the support set images, each represented with one sample (shot). Given a batch of tasks $\mathcal{T}_i$, the support set is first used to obtain task-specific model parameters $\theta_i'$ for each task by few gradient-step updates, which are then used together with the query set samples to perform a meta-update step for updating the meta-parameters $\theta$. After the meta-training is finished, for a new given task, the meta-trained model is used for inference by further adapting the meta-learned model with the support set, and measuring the performance on unseen query samples.

Larochelle, 2017; Finn et al., 2017; Raghu et al., 2019; Ye et al., 2020), meta-learning remains almost unexplored for multimodal few-shot settings. Similar to unimodal settings, empowering a multimodal few-shot learner with the ability to accrue knowledge across tasks, would assist in building internal multimodal representations broadly suitable for many tasks. These representations could serve as a bridge between the different modalities and assist in learning quickly new tasks by observing only limited labeled examples.

Motivated by this, we define a novel multimodal few-shot meta-learning approach, to bridge the gap between vision and language modalities, illustrated in Figure 1. Specifically, our approach adopts publicly available pre-trained large vision encoders (Radford et al., 2021) and language models (Brown et al., 2020), which are kept frozen to take advantages of their already-learned reasoning capability (Tsimpoukelli et al., 2021; Mokady et al., 2021; Zhou et al., 2022; Jia et al., 2022; Tewel et al., 2021; Zhai et al., 2021). In doing so, our method avoids the huge computational resources required by those models during training and their dependency on large datasets. Unlike prior multimodal few-shot learners (Tsimpoukelli et al., 2021; Alayrac et al., 2022), our approach decomposes the training of the model into sequential observing of multimodal few-shot tasks, in the spirit of meta-learning. During the meta-training stage, the model translates the visual representations into a visual prefix for the language model, by using a lightweight meta-mapper network. This network serves as a multimodal bridge between the large vision and language models and is entirely built from self-attention layers (Lee et al., 2019). Essentially, the aim of the meta-mapper is to collect shared meta-knowledge from related tasks, by mapping the visual representation into the latent space of the language model. Then, during inference, or meta-test according to the meta-learning parlance, the model is able to induce the task in a fully data-driven manner by observing few labeled examples and thus entirely removes the need for hand-engineered task inductions.

In summary, we contribute in three major aspects: *Conceptually*: We introduce meta-learning to multimodal few-shot learning, which enables fast adaptation and efficient learning of multimodal few-shot tasks. To that end, we design a new setting for multimodal few-shot learning by re-organizing existing datasets and following suitable benchmarks. *Methodologically*: We present a multimodal meta learner by using a lightweight meta-mapper, which learns to bridge a large frozen vision and language backbones. To the best of our knowledge, this is the first meta-learning based model to solve multimodal few-shot tasks. *Empirically*: We demonstrate through systematic experiments on those benchmarks that our model, using only the small trainable meta-mapper with frozen backbones, yields strong multimodal few-shot performance, while being computationally more efficient.

## 2 RELATED WORKS

**Large-scale language models** emerged since the introduction of attention (Bahdanau et al., 2015) and Transformers (Vaswani et al., 2017) to successfully deal with long-range dependencies in sequences. The field has seen significant progress over the last years (Devlin et al., 2019; Radford et al., 2019; Brown et al., 2020), which also initiated the development of similar strategies for vision (Dosovitskiy et al., 2021; Radford et al., 2021; Zhai et al., 2021) and multimodal models (Lu et al., 2019; Li et al., 2020; Wang et al., 2021; Jia et al., 2021; Tewel et al., 2021; Mokady et al., 2021; Eichenberg et al., 2021; Shen et al., 2021). In current few-shot scenarios, the language models are steered into producing a desired output by using the concept of prompting (Brown et al., 2020; Tsimpoukelli et al., 2021; Alayrac et al., 2022). The approach is to prepend fixed task induction with a few examples as a prompt and then generate the output from the language model. Instead of optimizing over a fixed set of task inductions and examples, also known as in-context learning, we aim to optimize the context representation (Li & Liang, 2021; Lester et al., 2021), in the form of meta-learned visual prefix.

**Meta-learning for few-shot learning** (Ravi & Larochelle, 2017; Santoro et al., 2016b; Finn et al., 2017; Vinyals et al., 2016; Snell et al., 2017) addresses the fundamental challenge of generalizing across tasks with limited labelled data. The aim is learning to acquire inductive biases and perform fast adaptation to new tasks (Grant et al., 2018). Meta-learning algorithms are typically categorized as: $(i)$ metric-based, focusing on learning a common embedding space and deriving prototypes as meta-knowledge (Vinyals et al., 2016; Snell et al., 2017; Sung et al., 2018) $(ii)$ memory-based, using an external memory as meta-knowledge to quickly adapt to new tasks (Mishra et al., 2018; Santoro et al., 2016a; Munkhdalai & Yu, 2017; Munkhdalai et al., 2018) and $(iii)$ optimization-based, aiming to learn a good model initialization across tasks as meta-knowledge, which can be used to efficiently adapt to new samples (Ravi & Larochelle, 2017; Finn et al., 2017; Grant et al., 2018). Our approach is positioned in this last category, as it offers greater flexibility and is modality-agnostic.

**Multimodal few-shot learning** across vision and language modalities has emerged recently with the introduction of the first multimodal few-shot learner Frozen (Tsimpoukelli et al., 2021), followed by other innovative prompting and in-context learning strategies (Song et al., 2022; Jin et al., 2021). Particularly, Frozen performs in-context learning, which is considered as one of the possible approaches to deal with multimodal few-shot learning settings. Another recently proposed model named Flamingo (Alayrac et al., 2022) follows a similar in-context learning paradigm, by using a visual-language model of 70 billion parameters, which is proven to be successful due to its scale. Our goals differ since our aim is to perform fast adaptation to new tasks by acquiring meta-knowledge across tasks, with a lightweight model of less than two million trainable parameters. In particular, we define optimization-based meta-learning steps (Ravi & Larochelle, 2017; Finn et al., 2017; Grant et al., 2018), by using the context samples to adapt a small trainable mapper and evaluate on the query samples.

## 3 METHODOLOGY

We aim to train a model that can learn and quickly adapt to new multimodal tasks with limited labeled data under the meta-learning setting. In the next sections, we will first define the multimodal meta-learning setting, then explain our architecture, and finally describe how it is used during training and inference time.

### 3.1 PROBLEM FORMULATION

We start by defining our approach for multimodal few-shot settings, following a meta-learning approach. Different from standard supervised learning, in meta-learning we are dealing with a collection of meta-datasets, which are split into disjoint $\mathcal{D}_{meta-train}$ and $\mathcal{D}_{meta-test}$ partitions. We define a meta-training stage, where the model is trained on the meta-train $\mathcal{D}_{meta-train}$ partition, and a separate meta-test stage where the performance is measured on the $\mathcal{D}_{meta-test}$ partition. In particular, the meta-train $\mathcal{D}_{meta-train}$ set consists of meta-datasets $\mathcal{D}_i$, each one containing a pair of separate inner-train set $\mathcal{D}_i^{tr}$, also referred to as a support set and inner-test set $\mathcal{D}_i^{ts}$, referred to as a query set. This means that $\mathcal{D}_{meta-train} = \{(\mathcal{D}_1^{tr}, \mathcal{D}_1^{ts}), \dots (\mathcal{D}_n^{tr}, \mathcal{D}_n^{ts})\}$. We refer to each meta-dataset pair $(D_i^{tr}, D_i^{ts})$ as a meta-task $\mathcal{T}_i$, following Finn et al. (2017).

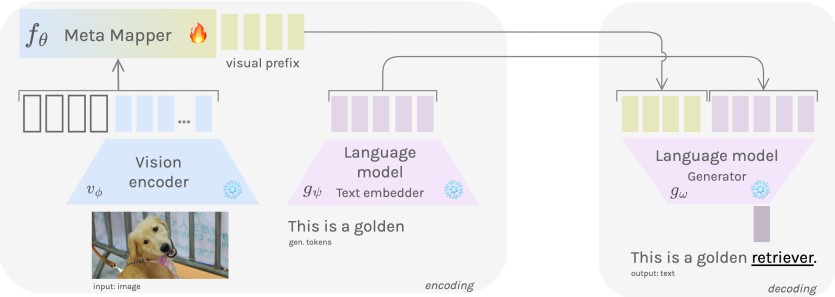

Figure 2: The architecture of the multimodal meta few-shot learner. It consists of three parts: frozen vision encoder $v_\phi$; frozen language model with a text embedder $g_\psi$ and a generator $g_\omega$; and a meta-mapper $f_\theta$ with trainable meta-parameters $\theta$. In the example shown, the model is generating the last word *retriever*, in an autoregressive manner.

When considering a $k$-shot, $N$-way setting, the support set $\mathcal{D}_i^{tr}$ of a single meta-task $\mathcal{T}_i$ consists of $k$ labeled samples for each of the $N$-ways, where $N$ is the number of object categories i.e. ways in few-shot parlance, meaning $\mathcal{D}_i^{tr} = \{(x_1^i, y_1^i) \dots (x_k^i, y_k^i)\}$, where $x_j^i$ represents the image and $y_j^i$ represents the caption. The query set $\mathcal{D}_i^{ts}$ is defined as $\mathcal{D}_i^{ts} = \{(x_1^i, q_1^i, a_1^i) \dots (x_m^i, q_m^i, a_m^i)\}$, where $x_j^i$ is the image, $q_m^i$ is an optional slot for an input text representation, like a question, and $a_m^i$ is the output text representation, i.e an answer to the question.

We consider a distribution over tasks $p(\mathcal{T})$, where a single batch of tasks $\mathcal{T}_i \sim p(\mathcal{T})$ for the $k$-shot $N$-way problem, is sampled in the following fashion: $N$ object categories are randomly sampled from the pool of all meta-train $\mathcal{D}_{meta-train}$ samples. For the support set there are $k$ samples chosen for each of the $N$ categories, and for the query set there are $m$ samples per category, with $m > k$.

## 3.2 MODEL ARCHITECTURE

The architecture that we present in this paper is modular and consists of three components: a vision encoder, the meta-mapper and a language model, illustrated in Figure 2. It is trained in a fully autoregressive manner, meaning that it offers flexibility for the choice of a downstream generative multimodal task.

**Vision encoder network**  The vision encoder is defined as a function $v_\phi$, with fixed parameters $\phi \in \mathbb{R}^{d_v}$, taken from a pre-trained vision encoder. The input is a raw image $\mathbf{x}$ and the outputs are the extracted visual features $v_\phi(\mathbf{x}) = x_1, \dots x_n$.

**Meta-mapper network**  To map the visual encoding into the latent space of the language model in our multimodal few-shot setting, we use a set of $l$ learnable parameters $p_i \in \mathbb{R}^{d_e}$. These parameters are also called visual prefix for the language model, with the same dimension $d_e$ as the language embeddings. We prepend this visual prefix to the encoded visual features, yielding the following sequence $[p_1 \dots p_l, x_1, \dots x_n]$. Particularly, we view this representation as an ordered set of elements and we employ self-attention to simultaneously encode the whole set. For this purpose, we adopt the set multi-head attention block (Lee et al., 2019), which has the role of a meta-mapper with trainable meta-parameters $\theta$, and is defined as:

$$\text{MetaMap}_\theta(Q, K, V) = \sigma(QK^T) * V, \tag{1}$$

where the pairwise dot-product $QK^T$ measures the similarity amongst features, and is used for feature weighting computed through an activation function $\sigma$. Intuitively, each feature of $V$ will get more weight if the dot-product between $Q$ and $K$ is larger. In Eq. 1, we have $Q = K = V = [p_1 \dots p_l, x_1, \dots x_n]$, meaning that the input of the meta-mapper is the sequence of features, and the output is a vector of learned parameters $p_1^* \dots p_k^*$, namely the visual prefix for the language model, as expressed in:

$$p_1^* \dots p_l^* = \text{MetaMap}_\theta([p_1 \dots p_l, x_1, \dots x_n]). \tag{2}$$

The self-attention layer, as the building block of this module, allows to retrieve meaningful information from the visual features $x_1, \dots x_n$, and accumulate it into $p_1^* \dots p_l^*$, due to its inherent weighting mechanism of pairwise similarities between elements in the sequence. The meta-parameters of the meta-mapper are learned and shared across all tasks $\mathcal{T}$ in $D_{meta-train}$.

**Language model**   The language model is defined as a function $g_\omega$, which parametrizes a probability distribution over a text sequence $\mathbf{y}$. It consists of an embedding function $g_\psi$, with $\psi \in \mathbb{R}^{d_e}$, to embed each word token $y_i$ of the sequence $\mathbf{y}$ into a word token embedding $t_i$, followed by a Transformer block conducting an autoregressive text generation. More precisely, the language model receives the visual prefix $p_1^* \ldots p_k^*$ concatenated with the token embeddings $t_1, \ldots t_m$, and outputs the next tokens conditioned on the prefix in an autoregressive manner:

$$t_{i+1} = g_\omega([p_1^* \ldots p_l^*, t_1, \ldots t_i]), i < m. \tag{3}$$

Since our approach aims to learn in a limited labeled space which might distort the learned parameters if we update them, we initialize the $\omega$ parameters with pre-trained parameters from a large-scale language model and keep them entirely frozen.

## 3.3   META-TRAINING & INFERENCE

To meta-train the model, we sample batches of multimodal few-shot learning tasks $\mathcal{T}_i \sim p(\mathcal{T})$, which consist of a support set denoted as $D_i^{tr}$ and a query set $D_i^{ts}$. Here, for simplicity, we assume that our full model described in 3.2 is defined as a function $f_\theta$, which receives an image $\mathbf{x}$ as input and produces $\mathbf{y}$ as output. The loss function, optimized per task during training, is a cross-entropy loss, defined as:

$$\mathcal{L}_{\mathcal{T}_i}(f_\theta) = \sum_{\mathbf{x}^j, \mathbf{y}^j \sim \mathcal{D}_i^{tr}} \mathbf{y}^j \log f_\theta(\mathbf{x}^j) + (1 - \mathbf{y}^j) \log(1 - f_\theta(\mathbf{x}^j)). \tag{4}$$

When adapting to a new task $\mathcal{T}_i$, the trainable meta parameters $\theta$ become *task-specific* parameters, namely $\theta_i'$. These task-specific parameters are computed with $N$ gradient-step updates, similar as in (Finn et al., 2017), with the following rule for one gradient update: $\theta_i' = \theta - \alpha \nabla_\theta \mathcal{L}_{\mathcal{T}_i}(f_\theta)$. This is referred as the inner-loop update, where $\alpha$ is the hyperparameter for the step size. Next, the model meta-parameters $\theta$ are optimized for the performance of $f_{\theta_i'}$, using the query set $D_i^{ts}$ samples and the task-specific parameters $\theta_i'$ as initialization of the model:

$$\min_\theta \sum_{\mathbf{x}^j, \mathbf{y}^j \sim \mathcal{D}_i^{ts}} \mathcal{L}_{\mathcal{T}_i}(f_{\theta_i'}) = \sum_{\mathbf{x}^j, \mathbf{y}^j \sim \mathcal{D}_i^{ts}} \mathcal{L}_{\mathcal{T}_i}(f_{\theta - \alpha \nabla_\theta \mathcal{L}_{\mathcal{T}_i}(f_\theta)}), \tag{5}$$

which is called outer-loop optimization. The meta-optimization across all tasks $\mathcal{T}_i$ is performed using stochastic gradient descent update rule, as follows: $\theta \leftarrow \theta - \beta \nabla_\theta \sum_{\mathbf{x}^j, \mathbf{y}^j \sim \mathcal{D}_i^{ts}} \mathcal{L}_{\mathcal{T}_i}(f_{\theta_i'})$, where $\beta$ is the step size hyperparameter.

In the meta-test stage, we consider new multimodal few-shot tasks $\mathcal{T}_i$ with previously unseen objects, which also have a support set: $\mathcal{D}_i^{tr}$ for fast adaptation by fine-tuning the model meta-parameters $\theta$ to a given task, and a query set $\mathcal{D}_i^{ts}$ to evaluate the model on this task. Note that the generation of the answer for each query set sample is done in an open-ended autoregressive manner. Specifically, we use top-$k$ nucleus sampling (Holtzman et al., 2019) to sample from the language model given the visual prefix of the image.

## 4   EXPERIMENTS

## 4.1   EXPERIMENTAL SETUP

**Datasets**   To design a meta-learning setting for multimodal few-shot learning, the datasets have to be structured into sequences of tasks, as explained in section 3.1. The datasets introduced in Frozen (Tsimpoukelli et al., 2021) are representative examples for this type of multimodal few-shot datasets, where each task consists of a few labeled instances as a support set and unlabeled ones as a query set. In practise, any dataset can be suited for few-shot meta-learning, as long as there is an available object information based on which the tasks can be constructed. Therefore, for meta-training, we use the COCO2017 captioning dataset (Lin et al., 2014) and restructure it to construct tasks in $N$-way, $k$-shot manner based on the $N$ object categories present in the images. For meta-test, we use the four mentioned datasets, namely, Real-Name miniImageNet, Real-Fast VQA, Open-Ended miniImageNet and Fast-VQA datasets, as they test the characteristics that make a good multimodal few-shot learner, such as fast binding of visual concepts to words and adaptation to new tasks.

Table 1: Comparison with the Frozen (Tsimpoukelli et al., 2021) baselines on Real-Name and Open-Ended miniImageNet 2- and 5-way setting; expressed in accuracy(%). ANIL (Raghu et al., 2019) is used as an upper bound, since it is a discriminative approach as opposed to our open-ended generative one. Our episodically trained models are outperforming the Frozen baselines, both with and without domain-shift. The overall best performance is denoted in bold, whereas the same settings as the baseline are denoted in italic & bold.

| Methods | episodic | cross-domain | Real-Name 2-way | | Open-Ended 2-way | | Real-Name 5-way | | Open-Ended 5-way | |
|---|---|---|---|---|---|---|---|---|---|---|
| | | | 1-shot | 5-shots | 1-shot | 5-shots | 1-shot | 5-shots | 1-shot | 5-shots |
| Frozen w/o task ind | ✗ | ✓ | 1.7 | - | 29.0 | - | 0.9 | - | 18.0 | - |
| Frozen w/ task ind | ✗ | ✓ | 33.7 | 66.0 | 53.4 | 58.9 | 14.5 | 33.8 | 20.2 | 21.3 |
| **Ours** | ✗ | ✗ | 35.6 | 65.7 | 50.2 | 57.5 | 15.2 | 39.6 | 18.9 | 22.0 |
| | ✗ | ✓ | *37.3* | *66.0* | *52.5* | *59.0* | *19.2* | *40.3* | *20.9* | *25.0* |
| | ✓ | ✓ | 45.3 | 69.8 | 53.6 | 63.4 | 24.7 | 41.8 | 24.8 | 28.5 |
| | ✓ | ✗ | **48.2** | **72.3** | **58.7** | **65.8** | **29.0** | **43.2** | **25.1** | **29.6** |
| ANIL upper-bound | - | - | 73.9 | 84.2 | - | - | 45.5 | 62.6 | - | - |

Table 2: Comparison with the Frozen (Tsimpoukelli et al., 2021) baselines on 2- and 5-way Real-Name and Open-Ended miniImageNet for the 1-shot setting, with different number of repeats (reps); expressed in accuracy(%). Increasing the repeats in the support set does not show a huge benefit compared to when the number of shots are increased. The overall best performance is denoted in bold, whereas the same settings as the baseline are denoted in italic & bold.

| Methods | episodic | cross-domain | Real-Name 2-way | | Open-Ended 2-way | | Real-Name 5-way | | Open-Ended 5-way | |
|---|---|---|---|---|---|---|---|---|---|---|
| | | | 1-rep | 5-reps | 1-rep | 5-reps | 1-rep | 5-reps | 1-rep | 5-reps |
| Frozen w/ task ind | ✗ | ✓ | **63.0** | 63.7 | 51.1 | 58.5 | **33.8** | 32.8 | 21.4 | 20.9 |
| **Ours** | ✗ | ✗ | 36.0 | 36.7 | 51.2 | 52.0 | 15.5 | 15.9 | 18.5 | 19.0 |
| | ✗ | ✓ | *37.9* | *38.5* | *53.0* | *54.2* | *20.3* | *22.5* | *25.6* | *26.1* |
| | ✓ | ✓ | 46.8 | 47.5 | 56.4 | 58.0 | 26.7 | 27.1 | 26.5 | 28.1 |
| | ✓ | ✗ | 62.1 | **64.3** | **59.8** | **60.8** | 31.3 | **33.7** | 26.8 | **27.5** |

**Training & inference procedures** We consider two different training procedures: *(i)* standard mini-batched training, which we refer to as a non-episodic; *(ii)* our proposed procedure by observing batches of multimodal tasks, which we refer to as episodic training; These training procedures are considered within two scenarios determined by the domain shift emerging during training and inference: *(i)* cross-domain multimodal few-shot learning, when the training and test datasets are of different distribution. Here the model is trained on COCO2017 (both in episodic and non-episodic manner) and the performance is measured on the full multimodal few-shot datasets; *(ii)* in-domain multimodal few-shot learning (standard meta-learning) setup, where the training and test partitions are of the same domain. This means that the training and test partitions, namely meta-training and meta-test tasks are constructed from the same multimodal few-shot dataset.

**Implementation details** The vision encoder is implemented as CLIP (Radford et al., 2021) with the Vision Transformer (ViT/B-32) (Dosovitskiy et al., 2021) as a backbone model, yielding visual features of size 512. The language model is implemented as GPT-2 (Radford et al., 2019), with word embeddings of size 768. The learnable visual prefix is empirically set to length 4 with dimensions of 768, which are specific to the language model. The meta-mapper is initialized following Xavier uniform initialization (Glorot & Bengio, 2010). For the meta-learning specific hyperparameters, we empirically determined to have five inner-loop gradient-updates with a learning rate of 0.01. The meta-update is performed with AdamW (Loshchilov & Hutter, 2018) with a meta-learning rate of 0.001 and 4 tasks in each meta-batch. All hyperparameters are tuned using the query sets in the meta-training partition which acts as a validation set.

Regarding the computational resources, our model is trained end-to-end on one NVIDIA GTX 1080Ti GPU, in less than 2 hours, showing the benefit of the lightweight framework. The total number of trainable parameters is approximately two million, which is order of magnitudes lower than the Frozen model. The full implementation is in PyTorch and HuggingFace (Wolf et al., 2020) and is publicly realized at `https://github.com/ivonajdenkoska/multimodal-meta-learn`.

Table 3: Comparison with the Frozen baseline (Tsimpoukelli et al., 2021) on 2-way Real-Fast VQA and Fast-VQA, in accuracy(%). Our episodically trained models outperform their counterparts, both with and without domain shift.

| Methods | episodic | cross-domain | Real-Fast VQA | | Fast-VQA | |
|---|---|---|---|---|---|---|
| | | | 1-shot | 5-shots | 1-shot | 5-shots |
| Frozen | ✗ | ✓ | 7.8 | 10.5 | 2.8 | 7.9 |
| **Ours** | ✗ | ✗ | 5.4 | 9.1 | 2.5 | 7.1 |
| | ✗ | ✓ | 6.9 | *10.7* | *3* | *8* |
| | ✓ | ✓ | 8.5 | 13 | 5.2 | 8.6 |
| | ✓ | ✗ | **9.7** | **13.2** | **5.7** | **9.3** |

Table 4: Benefit of the meta-knowledge in accuracy(%) on the four multimodal few-shot datasets under the 2-way setting. Accruing meta-knowledge is crucial for our approach.

| Variants | Real-Name | | Open-Ended | | Real-Fast VQA | | Fast-VQA | |
|---|---|---|---|---|---|---|---|---|
| | 1-shot | 5-shots | 1-shot | 5-shot | 1-shot | 5-shots | 1-shot | 5-shot |
| w/o meta-knowledge | 0.0 | 0.06 | 0.0 | 0.05 | 0.0 | 0.03 | 0.0 | 0.06 |
| **w/ meta-knowledge** | **48.2** | **72.3** | **58.7** | **65.8** | **9.7** | **13.2** | **5.7** | **9.3** |

## 4.2 RESULTS & DISCUSSION

**Binding of visual concepts and words**   The experiments conducted on Real-Name and Open-Ended miniImageNet measure to what extent the multimodal meta-learner is able to bind visual concepts to words. Table 1 shows the 2-way and 5-way accuracy in 1 and 5 shots on both datasets. We observe that our multimodal meta-learner is able to largely outperform Frozen, even without using an engineered task induction. This shows the advantage of having a meta-learned visual prefix, in contrast to just reshaping the vision encoder output as a prefix to language models. Specifically, the meta-learned prefix is able to collect shared meta-knowledge from related instances in the tasks, which is useful for narrowing down the search space in a learnable manner, instead of using an engineered task instruction. Having this adaptable component, makes it easier for the model to later adapt to unseen visual concepts, without being explicitly instructed to do so. Also, similar to Frozen, we observe an upward trend when increasing the number of shots, for both the 2-way and 5-way settings, which confirms that having more distinct examples of the task increases the performance.

We believe that the open-ended approach is promising due to its flexibility in reasoning about visual concepts in an unconstrained manner, instead of relying on a pre-defined closed set of concepts. However, due to the magnitudes larger hypothesis space in the open-ended text generation (the full vocabulary of the language model), compared to the finite set of possible answers of conventional classifiers, such as ANIL (Raghu et al., 2019), it is not possible to perform a fair comparison to them. Therefore, we use their results as upper bound to our approach.

**Increasing the number of repeats of support samples**   As an additional analysis, we experiment with the number of times each shot is repeated in the support set presented to the model, during inference i.e. the meta-test time. We observe from Table 2 that increasing the repeats in the support set, does not show as large benefit as increasing the number of distinct shots. This means that the meta-learner is able to accumulate more useful meta-knowledge by seeing different examples of images, even only once, rather than multiple repetitions of the same ones. Contrary to this observation, Frozen benefits more from these repetitions since they are used in combination with a hand-engineered task induction, which is crucial for their method.

**Visual question-answering with limited labeled samples**   The aim of the experiments on the Real-Fast and Fast-VQA 2-ways benchmarks, presented in Table 3 with 1 and 5 shots, is to evaluate the abilities of the multimodal meta-learner to answer more complex queries about the objects in the image. There is an implicit testing of the binding of visual concepts and words, since the query samples are designed in such a way to contain both categories from the support set in the query image, while the question is addressed in one of them. Moreover, the emphasis is on the ability of the meta-learner to reason about particular object properties, without being explicitly meta-trained to do so. As we observe from the results, our multimodal meta-learner achieves improvements over the

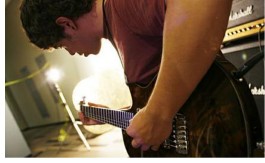
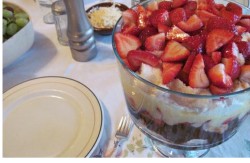
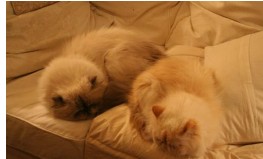
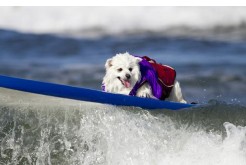

Q: This is a?
Ground-truth: This is an electric guitar.
Ours: This is an electric guitar player.

Q: This is a?
Ground-truth: This is a trifle.
Ours: This is a trifle.

Q: What breed of cat is this?
Ground-truth: Persian.
Ours: Persian breed.

Q: What does the dog ride?
Ground-truth: A blue surfboard.
Ours: A blue surfboard.

Figure 4: Qualitative examples of query set images from Real-Name miniImageNet (first two) and Real-Fast VQA (last two), with their question, ground-truth and answers generated by our model.

Frozen baseline, showing once more the benefit of accruing meta-knowledge across tasks. Again, there is an upward trend in the performance when increasing the number of shots.

**Episodic training of multimodal few-shot learners**   Another important observation from Tables 1 and 3 is that the episodic training improves the overall performance. The simple reason for this is having matched training and inference time conditions, as in standard supervised learning. More precisely, the multimodal few-shot datasets that are employed at inference time, are structured into tasks, so training the model in a similar manner by observing the tasks improves the performance.

**Learning across domains**   Tables 1 and 3 provide insights into the more challenging cross-domain multimodal few-shot setting. Essentially, meta-training on the COCO2017 captioning dataset, helps the model to bind visual concepts with richer semantic descriptions, compared to when training on the multimodal few-shot datasets (which have simpler captions). However, according to the evaluation procedure that we follow, also employed by Frozen, the model is expected to generate the exact word as the ground-truth, penalizing any paraphrasing of words. Therefore, a COCO2017 meta-trained model exhibits lower accuracy from a quantitative perspective when transferred to a miniImageNet-based dataset. However, in many cases the generated sentences are qualitatively better and more detailed, as we discuss in the next paragraph.

**Qualitative analysis**   In Figure 4, we show a examples of query images with the questions and answers at inference time, by the best version of our approach in Tables 1 and 3. The capability of the multimodal meta-learner to bind visual concepts to words is apparent: the model is able to connect the visual concepts in the image not only to *electric guitar*, as stated in the ground-truth, but also to the word *player*. This demonstrates that the model is correctly steered by the meta-learned visual prefix to leverage additional information about image, not necessarily present in the ground-truth. An important observation from these examples is that the generated answers have some differences w.r.t the ground-truth, which is however penalized during evaluation, since we count as correct only words that match, to be able to compare to Frozen. The evaluation should, however, measure whether the generated sentence matches the image in a more general manner,

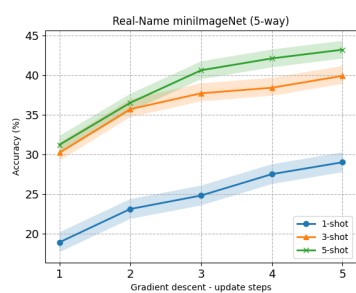

Figure 3: Relationship between consecutive steps of gradient-based updates and the accuracy on Real-Name miniImageNet on 5-way tasks, performed during the meta-test stage.

ideally without using ground-truth references (Hessel et al., 2021), which is left for future work.

### 4.3 ABLATION STUDY

**What is the benefit of accruing meta-knowledge?**   To test how useful the meta-knowledge is, we run an ablation experiment where the meta-knowledge accrued from the previous batch of tasks is erased before observing the next one. This essentially means to randomly initialize the meta-mapper, which collects the meta-knowledge, before observing the next batch of tasks. As it can be observed in Table 4, this erasing yields deteriorating performance, across all datasets, ways and shots. This shows that training the meta-mapper to accumulate useful representations from sequences of tasks is crucial for the success of our meta-approach.

Table 5: Benefit of the self-attention structure of the meta-mapper in accuracy (%) on the four multimodal few-shot datasets under the 2-way setting. Self-attention meta-mapper is more critical than the MLP-based one for our approach.

| | Real-Name | | Open-Ended | | Real-Fast VQA | | Fast-VQA | |
| Variants | 1-shot | 5-shots | 1-shot | 5-shot | 1-shot | 5-shots | 1-shot | 5-shot |
|---|---|---|---|---|---|---|---|---|
| MLP-based meta-mapper | 42.5 | 62.6 | 45.3 | 58.3 | 4.3 | 11.9 | 4.5 | 7.5 |
| **Self-attention meta-mapper** | **48.2** | **72.3** | **58.7** | **65.8** | **9.7** | **13.2** | **5.7** | **9.3** |

Table 6: Effect of using a fixed, hand-engineered task induction vs. only learning it from support sets, in accuracy (%) on the four multimodal few-shot datasets under the 2-way setting. Learning the task induction in a data-driven manner improves the performance.

| | Real-Name | | Open-Ended | | Real-Fast VQA | | Fast-VQA | |
| Variants | 1-shot | 5-shots | 1-shot | 5-shot | 1-shot | 5-shots | 1-shot | 5-shot |
|---|---|---|---|---|---|---|---|---|
| w/ fixed task ind | 47.9 | 71.9 | 58.0 | 65.0 | 9.5 | **13.2** | 5.0 | 8.9 |
| **w/o fixed task ind** | **48.2** | **72.3** | **58.7** | **65.8** | **9.7** | **13.2** | **5.7** | **9.3** |

**What is the effect of the multimodal inner loop optimization?** To analyse the effect of the gradient-based updates during test time, namely the inner loop optimization, we illustrate the relationship between the update steps and the accuracy in Figure 3. We observe increased performance as the number of gradient-based steps increases, for $\{1, 3, 5\}$ shots on 5-way Real-Name miniImageNet tasks. This demonstrates that the mechanism of the meta-learning framework to adapt the meta-parameters on support samples, allows the model to respond promptly to a new task, which yields increased performance on the query samples.

**Does the structure of the meta-mapper influence the performance?** To test the performance of our approach with a different structure of the meta-mapper, we run experiments with an alternative to the proposed self-attention-based one. Specifically, we implement a multi-layer perceptron (MLP) with one hidden layer. It can be observed from Table 5, that the self-attention meta-mapper brings notable improvements over MLP. This is attributed to the fact that, different from MLP, self-attention layers are able to select and extract the most relevant features from an input representation.

**Is there an effect of fixed task induction in the meta-learning setting?** We analyse the effect of a fixed hand-engineered task induction, as opposed to learning the induction in a data-driven manner. In particular, we choose the best variant of our meta-learner from Table 1, and add a fixed sentence describing the task in natural language, for instance, *"Answer with lion or dog."*, similar as in Frozen. It can be observed from Table 6, that there is no significant increase in the performance, meaning that only fine-tuning the meta-mapper on the support set is good enough to induce the task.

## 5 CONCLUSION

We present a novel meta-learning approach for multimodal few-shot learning. We design our model by introducing a lightweight trainable meta-mapper that acts as a bridge between large frozen vision and language models. This meta-mapper accrues shared meta-knowledge by sequentially observing tasks into a learnable visual prefix, which is then used to steer the language model into generating relevant linguistic interpretations. Unlike prior works, we use the support samples to adapt our model to unseen tasks, which shows to be beneficial as it improves performance without using hand-engineered task induction. Moreover, it operates with low computational costs, as it smartly combines large pre-trained models without the need to train them from scratch. This makes our architecture flexible enough to incorporate additional modalities, which is left to tackle in future work. Last but not least, our experiments verify the effectiveness of our approach by outperforming the baseline on several benchmarks and fostering further exploration of multimodal few-shot meta-learning.

## ETHICS STATEMENT

Our proposed model uses pre-trained large-scale models, both for the vision encoder and language decoder modules, namely CLIP (Radford et al., 2021) and GPT-2 (Radford et al., 2019). So, our model inherits any problematic biases and limitations that these pre-trained models may contain, meaning we should be careful when using the results especially in sensitive settings.

More specifically, our model has a generative nature with the ability to produce language, which might lead to generation of disinformation or potential hallucinations. But, as our model is conditional on the visual features, it enforces the text generation to follow the image contents. This serves as a control mechanism of the otherwise free text generation, and can mitigate the generation of linguistic disinformation. Additionally, as can be seen in some of the qualitative examples in Figure 4, our proposed approach takes advantage of the complementarities of the two backbone models. This suggests that it might be useful to mitigate biases potentially present in one of the pre-trained models.

## REPRODUCIBILITY STATEMENT

To help the reproducibility of the paper, we provide the formal definition of the meta-learning setup in Section 3.1, the implementation details in Section 4.1, additional details about the design of the setting and the formal definition of the algorithms during the meta-training and inferences stages in Appendix. Also, we release the source code to reproduce the results and evaluate the performance of the model at: `https://github.com/ivonajdenkoska/multimodal-meta-learn`.

## ACKNOWLEDGMENTS

This work is financially supported by the Inception Institute of Artificial Intelligence, the University of Amsterdam and the allowance Top consortia for Knowledge and Innovation (TKIs) from the Netherlands Ministry of Economic Affairs and Climate Policy.

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
