# OpenReview forum: "Meta Learning to Bridge Vision and Language Models for Multimodal Few-Shot Learning"
_ICLR.cc/2023/Conference — ICLR 2023 poster_

### Official Review · Reviewer_zhwM · 2022-10-25

**Confidence:** 4
**Correctness:** 3
**Technical Novelty And Significance:** 2
**Empirical Novelty And Significance:** 3
**Recommendation:** 3

**Clarity, Quality, Novelty And Reproducibility:**

Paper is very clear and nicely written - was a joy to read, but the originality is lacking.

**Strength And Weaknesses:**

Strengths:
1. The paper is well motivated and described. The ideas are clear and experiments are on several large multimodal datasets.
2. The paper is very clear with clear figures and exposition.
3. Experiments are comprehensive and results are strong.

Weaknesses:
1. The novelty of the paper is limited, since it seems to be a combination of Frozen (Tsimpoukelli et al., 2021) and standard meta-learning approaches to learn the mapping from image representations into language model token space.
2. There should be more comparisons in both performance and complexity - the proposed method does better but certainly uses more computation so the tradeoff should be analyzed.

**Summary Of The Paper:**

This paper proposes to combine meta learning with few-shot adaptation of language models, by essentially building upon methods such as Frozen (Tsimpoukelli et al., 2021) which maps image representations into language model token space, but adding a meta-mapper network, acting as a meta-learner, to more efficiently perform this mapping. The proposed approach can rapidly adapt to newly presented samples with only a few gradient updates, and shows strong results on recently proposed multimodal few-shot benchmarks.

**Summary Of The Review:**

The novelty of the paper is limited, since it seems to be a combination of Frozen (Tsimpoukelli et al., 2021) and standard meta-learning approaches to learn the mapping from image representations into language model token space.

---

> ### Author Response · Authors · 2022-11-17
> **Response to Reviewer zhwM**
>
> We thank the reviewer for the feedback and for acknowledging the ideas and expositions as clear. In the next paragraphs, we address the comments provided by the reviewer.
>
> **Novelty highlights**  We would like to elaborate on the novelty of this work in the following three major aspects, which we believe consolidate our contributions to multimodal few-shot learning:
> - **Conceptually**: To the best of our knowledge, this is the *first work* addressing multimodal few-shot learning from an optimization-based meta-learning perspective. To achieve that, we design a new meta-learning setting for multimodal few-shot learning with repurposed datasets, which follows the structure of few-shot settings in a natural way. The repurposed datasets based on COCO image captioning dataset for the cross-domain meta-training will be released as benchmarks to foster the research in this direction.
> - **Methodologically**: We design a novel architecture to bridge large-scale frozen vision and language models with a lightweight trainable meta-mapper. This meta-mapper is built from self-attention layers and is trained episodically to collect meta-knowledge across few-shot multimodal tasks. The meta-knowledge from related tasks is collected during training and used to induce the task by fine-tuning the model on unseen samples. This results in *fully data-driven* task inductions, avoiding to hand-engineer them, which is important for better generalization to novel unseen tasks.
> - **Empirically**: We have conducted a comprehensive set of experiments on 4 multimodal few-shot datasets, which demonstrate that our lightweight model achieves strong performance compared to its counterpart. Our experimental setup has not yet been entirely explored before in multimodal few-shot learning, since we embed meta-learning into multimodal settings. We report the performance for different training procedures, namely episodic and non-episodic, both on different/or the same domain, with a different number of ways, namely 2 and 5 - represented with different numbers of shots - 1 up to 5, which yields the following experimental setups:
>    - Episodically trained model for in-domain settings;
>    - Non-episodically trained for in-domain settings;
>    - Episodically trained model for cross-domain settings;
>    - Non-episodically trained for cross-domain settings.
>
>     In addition, we perform thorough ablation experiments to investigate the effect of our newly introduced components, both *conceptually* and *methodologically*, showing that all components contribute to the final result.
>
> **Performance and complexity w.r.t the baseline**
> The meta-training and inference stages in our approach are less computationally-expensive w.r.t the large-scale pre-training of the vision encoder of Frozen, while achieving comparable performance. To be more specific, Frozen pre-trains the vision encoder from scratch and then uses it to extract image features for prompting a frozen language model. By contrast, we freeze the backbones entirely and only train the meta-mapper. This results in a more *flexible* architecture that is also *less computationally* expensive and independent of the specific pre-training of large-scale models.
>
> During inference, we are fine-tuning the meta-mapper, by using the support set to adapt its parameters to the given task. This is different from Frozen, which performs direct prompting of the language model with no gradient-step updates. Although we perform a few gradient-step updates at inference time - due to the nature of the meta-learning algorithm - this adds a *minimal* complexity compared to training a whole vision encoder from scratch. For instance, the multimodal few-shot datasets that we use are orders of magnitude smaller than the pre-training dataset for Frozen. To be precise, the COCO image captioning dataset has 118k image-caption pairs as training data, whereas the Conceptual Captions used by Frozen for pre-training has approximately 3.3M image-caption pairs, which makes the training longer and more complex. Additionally, the baseline is training NF-ResNet-50 as a vision encoder, with the number of parameters in the order of 100M, whereas our trainable part has approximately 1.7M parameters. We thank the reviewer for this relevant comment, and we add this discussion in the revised version of the paper.
>
> Finally, we would like to mention that our work is *orthogonal* to those of large-scale vision and language models because we can always leverage their capacity, being independent of their pre-training objectives or datasets.

---

> > ### Comment · Reviewer_zhwM · 2022-11-30
> > **follow-up**
> >
> > Thank you authors for the detailed response and additional revisions to the paper. As I mentioned in my original review, the paper is well conceptualized and executed - the ideas are clear, and I appreciate the thorough experiments on several large multimodal datasets. The results are certainly excellent and the community would enjoy reading this paper. However, I still have the issue of novelty, which unfortunately is highly subjective. I am focusing on the conceptual technical contributions here (I acknowledge the strong empirical results). Can you answer the question: 'What is the specific new knowledge from this paper, that existing researchers in meta-learning or multimodal learning do not already know?'
> >
> > For example, we know that mapping from one modality (or more generally, domain) to another can be performed either be translation (x to y, all the work in modality translation and domain translation, and the Frozen paper) or contrastive learning (x,y pairs to 0/1 positive or negative, alignment models and CLIP etc). Folks have also showed that meta-learning can help learn this mapping quickly and for few-shot generalization: meta-learning for translating from one domain or modality to another: https://ojs.aaai.org/index.php/AAAI/article/view/6816, https://proceedings.neurips.cc/paper/2018/file/062ddb6c727310e76b6200b7c71f63b5-Paper.pdf (which does it for unsupervised pairs), and meta-learning for contrastive learning across modalities: https://arxiv.org/abs/2012.02813. There also seem to be other references such as https://arxiv.org/pdf/2109.13576.pdf (a survey paper, I believe the references related to meta-alignment are most relevant), https://ieeexplore.ieee.org/document/9054759, and https://ieeexplore.ieee.org/document/9514431 combining meta-learning and multimodal translation.
> >
> > I would be happy to increase my score if the authors can provide specific knowledge in applying meta-learning to multimodal translation, that is currently not known by the community. Is there a specific technical innovation, either to the meta-learning algorithm, or to the meta-parameters, when the 2 modalities are visual representations and language model embeddings?

---

> > > ### Author Response · Authors · 2022-12-08
> > > **Response to Reviewer zhwM (1)**
> > >
> > > We thank the Reviewer for the further engagement. We are happy to explain the specific knowledge we provided in our work and clarify the specific technical innovation.
> > >
> > > **The central specific knowledge** of this work is the data-driven task induction phenomena for multimodal few-shot settings - which we present in the paper. We observed that task induction in multimodal learning is crucial in scenarios with limited labeled data. Previous work, such as the Frozen baseline, adopted hand-engineered task induction, such as “Please answer with a lion or dog”, which was demonstrated to be key to their success. In contrast, our model is able to induce the task in a purely data-driven manner, by observing only a few demonstrations of the multimodal task during the meta-training stage. This specific way of meta-parameters optimization helps in leveraging meta-knowledge representations, which are useful when generalizing to unseen tasks as we show in this work. This finding has huge benefits, especially when it comes to more complex tasks since we would like the model to infer the task from the observed data, rather than having to always hand-engineer it.
> > >
> > > Then the specific technical innovation to meta-learning algorithms for multimodal few-shot learning stems from how we design the architecture to learn this task induction. We introduce the meta-mapper, which acts as a meta-learner built out of self-attention layers. This enables fast adaptation to a new multimodal task with only a few gradient-based updates of this trainable part. The meta-mapper is embedded in the encoder-decoder architecture and therefore seamlessly bridges the vision and language modalities.
> > >
> > > Additionally, our approach offers an alternative to in-context learning. Our method mitigates the shortcomings of in-context learning, which deals with the issue of fixed context window size of Transformer language models. Instead of using interleaved image-caption pairs as predefined context, in our framework we use them as a support set, to adapt the model to the current task, and afterward to measure the performance on its query set. Because of this flexible framework, our approach can be easily extended to having more shots in the support sets.
> > >
> > > **Regarding the mentioned references**, we thank the Reviewer for bringing these relevant references (enumerated below), which we will include for discussion in the future version for completeness. The contributions in our work are orthogonal to those in these papers. Here we would like to briefly explain the differences for your quick reference.
> > >
> > > The work presented in [1, 2] deals with the translation of one data modality across different domains. For instance, they learn how to map images from domain A to domain B or to transfer style from one image to another. Opposed to this, we strictly deal with translating from one data modality into another, by using generative language models and provide results about translating from different domains, w.r.t the distribution shift. This introduces new challenges in the translation process, as it is different from the mentioned work. In particular, we are trying to learn how to translate representations between *any* separately pre-trained large-scale backbones, without fine-tuning them. This approach has a huge potential for learning how to smartly merge large backbones, and reuse their already acquired knowledge in performing downstream tasks. We show that adopting alternative learning strategies, such as meta-learning, is worth considering, as a step towards leveraging large backbones, instead of repeatedly doing expensive pre-training or task-specific fine-tuning.
> > >
> > > Regarding the work presented in [3], they focus on different objectives, for instance, how large-scale image classification benchmarks can help in solving audio classification. This is *a different mechanism* of learning across modalities from the one we consider since we aim to translate from one modality to another in the scope of the same task (captioning, VQA). Additionally, [3] focuses on using separate encoders for different source and target modalities, whereas we focus on the complete encoder-decoder architectures to leverage the generation capabilities of large language models. Furthermore, the works presented in [4, 5] deal with a multi-task model that transfers knowledge from source tasks to target tasks, which can be essentially seen as a closed set prediction, similar to [1,2,3]. On the contrary, in our work, we deal with *open-ended predictions*, which highly differs and is more unconstrained than the closed one.
> > >
> > > *(continues in the next comment)*

---

> > > > ### Author Response · Authors · 2022-12-08
> > > > **Response to Reviewer zhwM (2)**
> > > >
> > > > Last but not least, the meta-alignment presented by other works summarized in [6], is one of the steps that our model achieves, but it is not our final objective. As we mentioned above, in our case, we deal with open-ended predictions and interpretation of images, which imposes additional challenges due to the free text generation of pure language models and the large variety of possible tokens and answers. Being able to bridge these large language models with other encoders, by observing only limited labeled data in a meta-learning manner, is a large step towards building more efficient and reusable multimodal architectures.
> > > >
> > > > We hope we managed to address the concerns of the Reviewer. We are of course happy to further elaborate on any part.
> > > >
> > > > **References**
> > > >
> > > > [1] Learning to Transfer: Unsupervised Domain Translation via Meta-Learning, Lin et al. (2020)
> > > >
> > > > [2] One-Shot Unsupervised Cross-Domain Translation, Benaim et al. (2018)
> > > >
> > > > [3] Cross-Modal Generalization: Learning in Low Resource Modalities via Meta-Alignment, Liang et al. (2020)
> > > >
> > > > [4] End-end Speech-to-text translation with modality agnostic meta-learning, Indurthi et al. (2020)
> > > >
> > > > [5] A Meta-Learning Approach for Fast Personalization of Modality Translation Models in Wearable Physiological Sensing, Akbari et al. (2022)
> > > >
> > > > [6] Multimodality in Meta-Learning: A Comprehensive Survey, Ma et al. (2022)

---

### Official Review · Reviewer_idxn · 2022-10-26

**Confidence:** 4
**Correctness:** 1
**Technical Novelty And Significance:** 3
**Empirical Novelty And Significance:** 3
**Recommendation:** 6

**Clarity, Quality, Novelty And Reproducibility:**

The overall writing quality is pretty clear, methods are well-motivated.

The overall quality of the paper is good, solid and comprehensive experiments, and the novelty is okay.

I believe the methods are reproducible.

**Strength And Weaknesses:**

Strength:

+ The main idea of combining soft-prompt tuning and MAML is very straightforward, and novel to some extent.
+ The empirical results are strong, and show that the proposed method outperforms the previous state-of-the-art in multimodal few-shot learning.
+ The ablation studies are quite comprehensive, and most of my questions about the methods was addressed.

Weakness:
- I don't see very obvious weakness of the methods.

Questions:

One question I have is about the variance of the model's performance, with respect to the few-shot support examples. Figure 3 has shown same hints but it could be interesting to more variance analysis on realistic VQA data.

**Summary Of The Paper:**

This paper propose a meta mapper network to tackle the task of multi-modal few-shot learning. Particularly, the proposed meta mapper contextualizes the soft visual prompts with the features of a pre-trained/frozen visual encoder and language models, for adapting the language model to output for the target task. During the inference, only the meta-mapper is fine-tuned via gradient based optimization, to adapt the model for the target multi-modal learning tasks. To train this meta-mapper, MAML-style of meta learning routine is employed for training. With this key idea, this paper achieves strong results on Real-Name mini-ImageNet and Open-Ended mini-ImageNet.

**Summary Of The Review:**

Overall I think this paper is making a solid contribution to the multi-model few-shot learning. The proposed method is interesting enough and effective.

---

> ### Author Response · Authors · 2022-11-17
> **Response to Reviewer idxn**
>
> We would like to thank the reviewer for the positive comments and feedback. In the next section, we address the question raised by the reviewer.
>
> **Variance of the model's performance**
> First, we would like to give some additional interpretation about the variance of the model’s performance in Figure 3. This figure illustrates the relationship between the accuracy and the gradient-based update steps in the inner loop, where we can notice a slightly larger variance. This is hinting at something which is relevant for any unimodal few-shot scenario: the magnitude of the variance in few-shot learning benchmarks largely depends on the total number of few-shot tasks in the meta-test stage, rather than the number of support set samples within a task. To be more specific, this means that if the few-shot datasets have more tasks to average over, naturally the variance in the reported accuracy will be smaller. For instance, in few-shot image classification datasets, such as MiniImageNet or Omniglot, the number of tasks is a lot larger compared to the multimodal few-shot datasets (Tsimpoukelli et al., 2021) that we consider in this paper,  thus the few-shot models trained on them exhibit smaller variance in their performance, which we also clarify in the revised version.
>
> To fulfill the request of the reviewer, we include similar figures for the VQA few-shot datasets, namely, Real-Fast VQA and Fast VQA datasets, showing this relationship between the performance in accuracy and the consecutive gradient-based update steps with 1, 3, and 5-shots settings. We observe similar phenomena in these figures w.r.t the variance of the model's performance, as in Figure 3. We provide a separate section in the appendix of the revised version of the paper to discuss these observations.

---

### Official Review · Reviewer_c2Z3 · 2022-10-30

**Confidence:** 5
**Correctness:** 4
**Technical Novelty And Significance:** 4
**Empirical Novelty And Significance:** 4
**Recommendation:** 8

**Clarity, Quality, Novelty And Reproducibility:**

Clarity
The paper is well written. It builds the approach logically based on a thorough literature survey and presents the proposed approach abd results clearly and concisely.
Quality and originality

The authors claim to be the first to apply meta-learning to multimodal few shot learning. That is a justified claim and the authors follow it up with good experimental results in both accuracy and ablation. I would therefore rate the quality and originality of this work highly.

**Details Of Ethics Concerns:**

It is unfortunate that the authors have neglected to address the ethical implications of their work directly. I am inclined to give them the benefit of the doubt given the quality of their work. They need to consider the following points:
1. It is well known that the large models that they are using as their backbones have various biases built into them. Therefore some caution is called for.
2. While keeping point 1 above in mind, it is also true that the proposed meta-learning approach bridges two modalities and in some qualitative results goes beyond the ground truth (The example with the electric guitar player comes to mind). My speculation is that the proposed meta-learning approach is ending up exploiting the complementarities of the two backbone models. That would imply that it will also help mitigate biases that are present in only one of the backbone models.
3. My suggestion to the authors would be to have a paragraph that directly addresses ethical concerns with the points 1 and 2 above in mind and any other points that they have.

**Strength And Weaknesses:**

Strengths:
1. Thorough and insightful literature survey that provides a sound motivation for the proposed approach.
2. Clear explanation of both the proposed approach and the experimental results.
3. Sound technical approach that is computationally lightweight and applicable to multiple multimodal tasks.
4. Experimental results show clear advance over the state of the art.
5. Insightful ablation results show the advantage of the proposed approach.

Weaknesses
1. No significant weaknesses. A suggestion - have you thought of how you would bridge three modalities? That is not required for this paper but I would be curious to know your thoughts. For example, how would you bridge visual, text and audio modalities? My guess is that your approach will scale up easily to accomodate more modalities but do think about it.

**Summary Of The Paper:**

The paper proposes a new approach to few-shot multimodal learning using a meta-learning approach that relies on generating visual prefixes that leverage prior training with other tasks. The visual prefix is produced by the proposed meta-mapper that meta-learns from previous tasks. The prefix approach enables seamless bridging of the textual and visual modalities. The proposed approach is computationally lightweight since it uses frozen backbone visual and text encoders (i.e. does not require retraining those), and clearly advances over the state of the art in its results. The paper also presents several insightful ablation results that establish the significance of the proposed approach to meta-learning.

**Summary Of The Review:**

Well written paper. Clear advancement of the state of the art with both the approach and experimental results. Insightful results from ablation. Good paper.

---

> ### Author Response · Authors · 2022-11-17
> **Response to Reviewer c2Z3**
>
> We would like to thank the reviewer for the positive rating and useful feedback. Below we provide our discussion about the points raised by the reviewer.
>
> **Bridging of three modalities** The comment by the reviewer gives valuable hints about the future work of this research direction. Indeed, our method has a flexible architecture and can accommodate any encoder or decoder networks, as well as any other modality. The higher-level goal of our method is to communicate multimodal concepts to a language model in the shape of latent prefixes - all mapped in the latent space of the language model. Adding one more modality will be straightforward -  for instance, we would use an encoder to generate the audio-specific features, which will be used as input to the meta-mapper yielding an audio prefix for the language model. Since the objective is to map the different modalities into representations understandable for the language model, we could easily concatenate all mapped representations into a single sequence and use it as a prompt to the language model. While being theoretically possible to design such architecture, implementing and training such a model in practice would require having appropriate datasets of image-text-audio triplets and also pre-trained audio encoders. We added this point as future work in the conclusion.
>
> **Ethical implications** Regarding the ethical implications of our work, we thank the reviewer for the suggestions and we agree that it is relevant to discuss it in the paper. We followed the advice and added the following paragraph as the “Ethics statement” of the paper:
> - Our proposed model uses pre-trained large-scale models, both for the vision encoder and language decoder modules, namely CLIP (Radford et al. 2021) and GPT-2 (Brown et al. 2020). So, our model inherits any problematic biases and limitations that these pre-trained models may contain, meaning we should be careful when using the results, especially in sensitive settings. More specifically, our model has a generative nature with the ability to produce language, which might lead to generation of disinformation or potential hallucinations. But, as our model is conditional on the visual features, it enforces the text generation to follow the image contents. This serves as a control mechanism of the otherwise free text generation and can mitigate the generation of linguistic disinformation. Additionally, as can be seen in some of the qualitative examples in Figure 4, our proposed approach takes advantage of the complementarities of the two backbone models, which might be useful for mitigating biases, potentially present in one of the pre-trained models.

---

### Official Review · Reviewer_VtRH · 2022-10-30

**Confidence:** 4
**Correctness:** 3
**Technical Novelty And Significance:** 3
**Empirical Novelty And Significance:** 2
**Recommendation:** 6

**Clarity, Quality, Novelty And Reproducibility:**

This paper is very well written and is very easy to understand. The frozen backbone idea itself has been explored by a few existing methods, e.g. freezing the language backbone (Frozen). Adopting meta-learning to bridge both vision and language backbone is original, and it also works nicely on the benchmarks. This paper has not attached/released the code, which makes it non-trivial to be reproduced.


**Strength And Weaknesses:**

Strength:
* The idea itself is very simple and easy to understand, for an impactful research direction. Making use of already pre-trained vision/language models could help save computational cost.
* This paper is very well written and is very easy to understand. The illustrations of the idea are very clear.
* Good qualitative results and ablation tests.

Weaknesses:
* It is appreciated to mark both "episodic" and "cross-domain" in Table 1 and Table 2. It is clear that the proposed "episodic" training helps on the quality significantly on target tasks. However, "cross-domain" leads to unfair comparisons, where the Frozen baseline only reported results with "cross-domain" setup while this paper highlights results with non "cross-domain" ("in-domain") setups. I am concerned by this point and if we look at "episodic" and "cross-domain" row in Table 2, this paper falls behind Frozen overall.


**Summary Of The Paper:**

This paper proposed to use meta learning for multimodal few-shot learning. Unlike previous methods that may train fully or partially the vision or language models, this paper freezes both vision and language models while only training the learnable parameters for the meta-mapper. Thanks to the frozen backbones, the meta-learning process is fast and able to make use of the pre-trained knowledge. The experimental results show that the proposed method outperformed the existing Frozen multimodal learning baseline.


**Summary Of The Review:**

This paper proposed a simple idea to bridge pre-trained vision and language backbones via meta-learning. The training is parameter efficient and fast thanks to frozen backbones. Concerns regarding the "cross-domain" comparisons lead to borderline rating.

---

> ### Author Response · Authors · 2022-11-17
> **Response to Reviewer VtRH**
>
> We would like to thank the reviewer for the positive comments and the acknowledgement of the research direction as impactful. Below we will address all points raised by the reviewer.
>
> **Clarification of the cross-domain results**
> Regarding the comparisons to the Frozen baseline, w.r.t the cross-domain and in-domain settings, we would like to emphasize that our experiments show better performance than Frozen in both settings - in Table 1. We agree and take the reviewer's advice to highlight the difference, by using italics and bold in the revised paper. Concerning the in-domain setup, our model shows higher performance compared to the cross-domain one, as commonly observed when the training and test partitions come from the same distribution. Frozen is not considering this in-domain setting, nor has released the pre-trained model or code, thus we could not include such results in our tables. We consider the in-domain setting as a relevant one in any few-shot settings, thus we incorporate it in our experimental design. Additionally, we add the results for all settings in terms of different numbers of shots, including the 3 shots in the appendix, to demonstrate the steady increase in performance when the number of shots increases.
>
> For the experiments investigating different numbers of repeats in Table 2, as the reviewer mentions, our model sometimes scores lower compared to the Frozen baseline for cross-domain settings. This can be expected as the Frozen model is using a hand-engineered task induction, such as *“Answer with lion or dog”*, in combination with these repetitions, which is crucial for the success of their method. Additionally, using this kind of fixed induction eases the text generation task, since it can be seen as “leaking” the correct answer to the language model in its prompt. In our work, the aim is to avoid using hand-engineered sentences, as we *learn* the task induction in a meta-learning manner, which favors seeing distinctive samples rather than repetitions.
>
> Moreover, this number of repetitions in Frozen is a hyperparameter, which is crucial and needs to be prefixed manually. By contrast, our approach does not rely on this kind of hyperparameter. Furthermore, we observe in the cross-domain performance of Frozen that increasing repetitions *does not* consistently improve their accuracy across different ways and shots, which makes it doubtful to conclude if repetitions are indeed helpful. We thank the reviewer for this useful remark, and we added these points in the revised version accordingly.
>
> **Code availability** We mentioned in the original submission that our code will be publicly released. Also, we added a reproducibility statement in the revised version of the paper, stating the following:
> “To help the reproducibility of the paper, we provide the formal definition of the meta-learning setup in Section 3.1, detailed description of the architecture in Section 3.2 and the implementation details in Section 4.1. Also, we provide additional details about the design of the setting and the formal definition of the algorithms during the meta-training and inferences stages in Appendix C. Also, we release the source code to reproduce the results and evaluate the performance of the model at: https://github.com/<redacted>/<redacted>.

---

### Author Response · Authors · 2022-11-17
**Thank you to all Reviewers for their feedback and acknowledgment**

We would like to thank all reviewers for their constructive feedback and acknowledgment of the paper. The questions raised by the reviewers are addressed separately in each comment section. The main points addressed below and in the revised version of the paper are:
- Clarification of the cross-domain setup and explanation about the influence of repetitions of shots (Reviewer VtRH);
- Reproducibility statement and clarification about the code availability (Reviewer VtRH);
- Discussion about incorporating a third modality, such as audio (Reviewer c2Z3);
- Discussion about the ethical implications of our work (Reviewer c2Z3);
- Additional interpretation of the variance of the model's performance (Reviewer idxn);
- Additional figures, similar to Figure 3, for the VQA datasets (Reviewer idxn);
- Highlighting the novelty and contributions (Reviewer zhwM);
- Discussion about the performance and complexity trade-off (Reviewer zhwM).

We are happy to continue the discussion and to further address any questions or requests the reviewers might have.

---

### Author Response · Authors · 2022-11-29
**Thank you to all Reviewers**

Dear Reviewers,

Thank you for your time and efforts in reviewing our paper. We would highly appreciate it if you could go over our responses, and please let us know if you have any further concerns or suggestions about our work.

The Authors

---

### Decision · Program_Chairs · 2023-01-20

**Decision:**

Accept: poster

**Justification For Why Not Higher Score:**

If we were to pass on the most negative review (see argument above) and look at the positive case, the paper is decently well supported by the reviewers. I don't think there's enough enthusiasm to carry this to a spotlight, but wouldn't object if it was bumped up.

**Justification For Why Not Lower Score:**

See review summary.

**Metareview: Summary, Strengths And Weaknesses:**

On the basis of the average score, this may seem like a borderline paper, however barring the review of Reviewer zhwM (score 3) the remaining reviews range from sympathy to outright support for publication. Focussing on Reviewer zhwM's comments, it seems the main concern there is the issue of novelty, with the reviewer contending that the proposed work resembled the method from Frozen (Tsimpoukelli et al., 2021) applied to a different domain (which some might argue, were this the case, that this is novel). The reviewer gave a detailed review, replied to most author responses, was cordial and not hostile, so I do not wish to completely discount their criticism. My main reasons for weighing it less in my decision are:
* Although an argument for novelty can be made, such matters are highly subjective, as the reviewer points out. The rest of their comments about the paper are actually quite positive, and so while I would feel comfortable seeing a score of, say, 5 associated with this review, a 3 seems low.
* Furthermore, I think the authors have responded to the points the reviewer brings up, especially regarding the conditions under which they would raise their score. Unfortunately, the reviewer was not able to reply to these final comments, so it is not known whether or not they would indeed raise their score, but from reading the response myself I think the benefit of doubt leans towards assuming they would temper the depth or confidence of their negative feedback.

As such, on the balance of the reviews and arguments made, I think this paper is entirely acceptable to the conference.

**Note From Pc:**

if the above contains the word "oral" or "spotlight" please see: "oral" presentation means -> notable-top-5% and "spotlight" means -> notable-top-25%. As stated in our emails, we are disassociating presentation type from AC recommendations